# Vasomotor Dysfunction in Patients with Ischemia and Non-Obstructive Coronary Artery Disease: Current Diagnostic and Therapeutic Strategies

**DOI:** 10.3390/biomedicines9121774

**Published:** 2021-11-26

**Authors:** Amr Abouelnour, Tommaso Gori

**Affiliations:** 1Zentrum für Kardiologie, Kardiologie I, und Deutsches Zentrum für Herz und Kreislauf Forschung, University Medical Center Mainz, 55131 Standort Rhein-Main, Germany; aelbadry@au.edu.eg; 2Cardiovascular Institute, Assiut University, Assiut 71515, Egypt

**Keywords:** INOCA, ANOCA, microvascular angina, coronary microvascular dysfunction, vasospastic angina, diagnosis, treatment

## Abstract

Many patients who present with symptoms or objective evidence of ischemia have no or non-physiologically-significant disease on invasive coronary angiography. The diagnosis of ischemic heart disease is thus often dismissed, and patients receive false reassurance or other diagnoses are pursued. We now know that a significant proportion of these patients have coronary microvascular dysfunction and/or vasospastic disease as the underlying pathophysiology of their clinical presentation. Making the correct diagnosis of such abnormalities is important not only because they impact the quality of life, with recurring symptoms and unnecessary repeated testing, but also because they increase the risk for adverse cardiovascular events. The mainstay of diagnosis remains an invasive comprehensive physiologic assessment, which further allows stratifying these patients into appropriate “endotypes”. It has been shown that tailoring treatment to the patient’s assigned endotype improves symptoms and quality of life. In addition to the conventional drugs used in chronic stable angina, multiple newer agents are being investigated. Moreover, innovative non-pharmacologic and interventional therapies are emerging to provide a bail-out in refractory cases. Many of these novel therapies fail to show consistent benefits, but others show quite promising results.

## 1. Introduction

The most commonly recognized and treated pathophysiological mechanism underlying ischemic heart disease is obstructive coronary artery disease (CAD). However, it was shown that only slightly more than one-third of patients with suspected stable CAD prove to have epicardial obstructive disease on invasive coronary angiography (CAG) [1]. Moreover, women with stable angina seem twice as likely to have normal coronaries or diffuse non-obstructive CAD, when referred for CAG [2]. Historically, the term “Cardiac Syndrome X” has been used to refer to patients with angina, ECG results demonstrating ischemia, and smooth coronary arteries on angiography [3]. We now know that there is a broad range of pathophysiological abnormalities that are not all captured with conventional exercise testing, and cannot be grouped under one single definition [4].

The pursuit of obstructive CAD could start with an anatomic strategy, i.e., computed tomography (CT) or invasive CAG, hence the term ANOCA (angina with non-obstructive coronary arteries) [5], or a functional strategy (stress testing) to seize objective evidence of ischemia, hence the term INOCA (Ischemia with non-obstructive coronary arteries) [6]. In other words, the terms ANOCA and INOCA have emerged to describe the most commonly encountered patient group presenting with angina and/or ischemia on stress testing, but no significant narrowing in their coronary arteries on anatomic testing. In such cases, the symptoms could be attributable to vasospasm, microcirculatory dysfunction, myocardial bridging, or even non-coronary etiologies.

It is now established that ANOCA/INOCA are not benign conditions [7]. Not only do they degrade the quality of life due to persistent symptoms, repeated hospitalizations, and unnecessary angiographies [8,9], but they are also associated with increased risk of major adverse cardiovascular events (MACE, mostly inappropriate revascularization but also myocardial infarction and death) [2,10], particularly in those with documented unequivocal myocardial ischemia [11].

### Search Methodology

We conducted a systematic review of published studies. The search was implemented in September 2021 in Pubmed using the following strategy:For the general description of microvascular disease, the following search syntax was used—coronary AND (“Microvascular angina” OR “syndrome X” OR “microvascular disease” OR “microvascular dysfunction”), and this yielded 3571 results;For different therapeutic/pharmacological approaches, the following search syntax was used—coronary AND (“Microvascular angina” OR “syndrome X” OR “microvascular disease” OR “microvascular dysfunction” OR “coronary flow reserve” OR “index of microvascular resistance *”) AND (therapy OR pharm *) AND prospective, and this yielded 353 results;For the prognosis/outcome, the following search syntax was used—(“Microvascular angina” OR “syndrome X” OR “microvascular disease” or “microvascular dysfunction”) AND (coronary) AND (“follow-up” OR hospitalization OR MACE OR death OR intervention OR “major adverse events” OR Follow * OR Predictor * OR Outcome * OR “myocardial infarction *” OR stroke * OR “heart attack *” OR hospitalization* OR “heart failure”), and this yielded 2866 results, which were already included in the results yielded by the first syntax.

Articles were restricted to the English language and published literature. Both authors screened the retrieved citations, and the decision to include a manuscript was based on agreement. A total of 182 articles were finally incorporated (including relevant cross-referenced articles), after removing duplicates.

## 2. Pathophysiology and Endotypes

Key to successful diagnosis is the understanding of the multiple potential “endotypes” [12], which are the distinct underlying pathophysiological mechanisms of ischemia and/or symptoms, rather than clustering all patients under the presenting of a “phenotype” of angina or angina equivalent. In INOCA, ischemia may be caused by any of the following mechanisms, listed below, which may coexist.

### 2.1. Coronary Microvascular Dysfunction (CMD)/Microvascular Angina (MVA)

This includes one or both of the following.

Fixed structural remodeling of the microvasculature, with reduced microcirculatory conductance (arteriolar intimal thickening, smooth muscle cell proliferation, perivascular fibrosis, as well as capillary rarefaction), which may occur in association with traditional risk factors for atherosclerosis, left ventricular hypertrophy, or cardiomyopathies [13]. The hemodynamic profile would be a reduced coronary flow reserve (CFR < 2) and an increase in microcirculatory resistance, as assessed, for instance by the index of microvascular resistances (IMR, normal considered to be < 25), during hyperemia induced by a non-endothelium-dependent vasodilator, such as adenosine, papaverine or regadenoson [12]. Patients with impaired CFR but normal IMR are usually (and inappropriately) included in this group by current expert opinion and guidelines [12]. These patients remain incompletely understood and require further critical assessment. Since CFR is a ratio of hyperemic to resting blood flow, a blunted CFR might be an expression of both decreased maximal flow (due, for instance, to microvascular disease) or high resting flow (which is common is conditions not necessarily associated with microvascular disease, such as hypertension or anemia). Additionally, another phenotype, previously described as “syn-drome Y” for the possible involvement of an inappropriately high bioavailability of the vasoconstrictor substance Y, is characterized by increased resistances at rest with normal reactivity to adenosine [14]. Typical of this phenotype is the evidence of slow coronary flow at angiography.

Dynamic arteriolar obstruction with vasomotor dysregulation of the coro-nary arterioles [15], whereby the presence of endothelial dysfunction is associated with impaired vasodilation and even paradoxical arteriolar vasoconstriction, that occur when myocardial oxygen demand increases [16]. This results in a corresponding hemodynamic pattern, observed during a vasoreactivity (acetylcholine (ACh) or ergonovine) challenge, with a limited increase or even a marked reduction in the blood flow, equivalent to the no-reflow phenomenon, in the absence of focal tight epicardial vessel spasm, signifying arteriolar spasm (although a diffuse narrowing of the distal epicardial vessels can develop) [12]. In these patients, CFR and IMR under non-endothelium-dependent vasodilators are normal, while microvascular resistance increases in response to ACh. This microvascular spasm is generally not quantified, but just diagnosed as typical angina and ST-changes on the ECG.

### 2.2. Epicardial Vasospastic Angina (VSA)

Originally described by Prinzmetal et al. as a “variant form” of angina pectoris, with ST elevation or sometimes “spurious improvement” during attacks [17], it is believed to be caused by a temporary and cyclic increase in the tonus of a narrowed vessel. Descriptions of other clinical forms later ensued [12].

The mechanism of such vasospasm was shown by experimental and clinical studies to be the hyper-reactivity of vascular smooth muscle cells to several potential constrictor stimuli. Endothelial dysfunction seems to facilitate this process, rather than being its direct cause [18].

A schematic presentation of the different phenotypes is presented in Table 1.

## 3. Risk Factors

Men have been shown to have a comparable prevalence of coronary vascular dysfunction to women. However, they probably have a higher prevalence of epicardial spasm and a lower prevalence of microvascular spasm than women [19].

### 3.1. CMD

Traditional risk factors for obstructive CAD are not consistently associated with A/INOCA [20]. Whereas some studies report an association of age, diabetes, hypertension, dyslipidemia, and central obesity, with microvascular disease, others report poor correlations [21,22,23,24,25]. Nevertheless, smoking is associated with impaired endothelium-dependent coronary vasodilation [26].

CMD contributes to angina in patients with SLE (Systemic Lupus Erythematosus) and rheumatoid arthritis with no obstructive coronary disease [27,28]. Similarly, CMD is reported to predict cardiovascular adverse events in psoriasis patients [29]. An association of CMD with inflammatory bowel disease has been reported as well [30].

### 3.2. Vasospastic Angina

Asians (Japanese and Taiwanese) people seem to be more at risk than Caucasians [31,32]. Smoking is a significant risk factor as well [33].

## 4. Symptoms

Similar to obstructive CAD, INOCA mainly presents with angina, but can also present with angina equivalents, i.e., dyspnea, fatigue, palpitations, indigestion, nausea, sleep disturbances, etc. [12,34].

Microvascular angina may be difficult to distinguish from that caused by obstructive CAD as both are primarily exercise-related. However, microvascular angina more frequently persists for several minutes after exercise, and can be poorly responsive to nitrates. It also occurs more often after exercise and is more often triggered by mental stress. Extreme tiredness may occur after an angina episode [12,34].

On the other hand, pure vasospasm seems to be less common, and is characterized by angina at rest, particularly between night and early morning [34], although in one study a circadian pattern of symptoms that was notably similar to non-vasospastic ANOCA patients was observed [35]. One fact that is peculiar to INOCA is that clinical symptoms can vary widely over time.

## 5. Diagnostic Approaches

The diagnostic gold standard for INOCA endotypes remains the invasive combined pressure-flow assessment. In addition, only invasive assessment can probe endothelial-dependent vasoreactivity to establish a diagnosis of vasospastic angina. Although the term INOCA implies the absence of a significant epicardial CAD, it should be noted that coronary vascular dysfunction can coexist with concomitant obstructive CAD [12,22].

As the coronary microcirculation is too small to be directly imaged in vivo, it can only be evaluated indirectly by its perturbed function when diseased.

In order to streamline diagnostic and therapeutic efforts, the Coronary Vasomotor Disorders International Study (COVADIS) group issued standardized criteria for diagnosing coronary microvascular angina (MVA) [36], as well as vasospastic angina [37].

### 5.1. Non-Invasive Diagnosis

Non-invasive techniques are able to detect regional differences in myocardial perfusion and/or wall motion, and thereby assign ischemia to epicardial territories. However, the more diffuse myocardial involvement in CMD (balanced ischemia) is relatively more challenging to detect. Therefore, quantitative rather than qualitative assessment is needed.

To address all endotypes of INOCA, a comprehensive assessment of the coronary function is needed, including both microvascular dysfunction and coronary spasm. An important limitation of non-invasive techniques is that they can only assess CFR, and not vasospasm. Although endothelial-dependent vasoreactivity can be assessed using a cold pressor test, there is only moderate correlation to invasive ACh vasoreactivity testing, and thus this is not an adequate alternative [38].

Moreover, non-invasive tests can have limited accuracy for predicting endothelium-independent dysfunction (e.g., as assessed by adenosine) [39]. No consensus is available so far on a universal cut-off across non-invasive imaging modalities.

#### 5.1.1. Transthoracic Echocardiography (TTE)

TTE allows for measuring coronary flow velocity reserve by measuring Doppler flow signals at rest and during induced hyperemia. This is usually done by sampling the left anterior descending artery (LAD) distally [40].

Adenosine or dipyridamole are usually used to induce hyperemia, which is only partially endothelial-dependent. In contrast, cold pressor testing utilizes sympathetic stimulation to induce exclusively endothelial-dependent hyperemia [41].

Despite being widely available, radiation-free, and relatively inexpensive, this modality carries multiple disadvantages. The assessment is technically demanding with numerous possible pitfalls, e.g., inadvertent mapping of different coronary tracts during the same session, or of wall noise, or right ventricular flow. Furthermore, the right coronary artery and to a greater extent the left circumflex arteries are quite challenging to sample, as compared to the LAD. That said, a TTE-derived CFR of ≤ 2.0 was shown to hold prognostic value [42,43,44], regardless of the subject being on anti-ischemic therapy at the time of testing [45].

#### 5.1.2. Positron Emission Tomography (PET)

Dynamic PET myocardial imaging is the reference standard for the non-invasive evaluation of myocardial blood flow (MBF) and subsequently CFR.

The quantification of absolute resting and hyperemic MBF in absolute units, e.g., mL/min/gm tissue (and myocardial flow reserve (MFR)/CFR (representing a ratio of the two)) is a strength of PET perfusion imaging [46]. These values are derived from dynamic acquisition with measurements of resultant myocardial and blood pool time–activity curves. PET has been largely validated against invasive tests, as well as in preclinical animal models by microsphere blood flow studies, for the absolute quantification of MBF and CFR, and has been shown to be highly reproducible over flows ranging widely from 0.5 to 6 mL/g/min [47].

Compared with SPECT, PET offers enhanced image quality because of higher count rates, better spatial resolution, and robust attenuation correction. Furthermore, PET tracers such as ^15^O-water and ^13^N-ammonia track MBF better at higher values compared to ^99m^Tc-labeled SPECT tracers [48].

It is important to recognize that various forms of stress, different radiotracers, and characteristics of particular software packages (unless using the same kinetic modeling) can all influence derived quantitative values. In addition, various “ischemic cut-offs” have been reported for absolute hyperemic flow as well as CFR, due at least in part to the different comparator reference standards used, e.g., perfusion defects with clinical angina and/or significant ECG changes during stress, versus an FFR ≤ 0.8, etc. [49].

It is, however, generally accepted by the American Society of Nuclear Cardiology (ASNC) and the Society of Nuclear Medicine and Molecular Imaging (SNMMI) that MFR > 2.3 indicates a favorable prognosis (assuming no lower regional value), whereas MFR < 1.5 (in the absence of concomitant elevated resting flow) is associated with high cardiac risk [50].

Absolute MBF quantification is well established with ^13^N-ammonia and ^15^O-water, but is more challenging with ^82^Rb because of its 75 s half-life resulting in noisy myocardial and blood pool time–activity curves [50]. Of note, the use of PET is limited by its availability, high cost, relatively challenging logistics, and exposure to radiation (albeit usually less than SPECT because of the shorter tracer half-life) [51].

#### 5.1.3. Cardiac Magnetic Resonance (CMR)

CMR offers an alternative to PET with the advantage of no ionizing radiation exposure, providing comprehensive data on ventricular size, function, and scarring within a single study. However, its wider adoption is hampered in part by the time-consuming offline processing and other technical issues described below [52].

CMR relies on measuring the myocardial perfusion reserve index (MPRI), which is equivalent to the MFR on PET. To quantify MBF at rest and stress, different modeling techniques can be used. A recent study performing head-to-head comparison of four different mathematical modeling applications concluded that there was no difference in diagnostic performance [53]. Another reported technique uses velocity-encoded imaging to derive CFR, by comparing through-plane flow in the coronary sinus at hyperemia to that at baseline [54].

A challenge in MBF quantification by CMR is the lack of linearity between the measured signal and contrast agent concentration [48]. To overcome such a challenge, a free-breathing motion-corrected dual sequence with separate pulse sequences for the arterial input function (AIF) and myocardial tissue has been optimized [55]. With fully automated in-line perfusion mapping, pixel-wise quantification of MBF was possible with good intra-study and inter-study repeatability at both rest and stress, albeit on healthy subjects [52]. This new optimized sequence was validated against ^13^N-NH3 PET in patients with stable CAD, and showed good agreement, promising to be a viable alternative [56]. Using the same optimized acquisition and reconstruction, another study proposed a stress MBF value of ≤1.94 mL/g/min to accurately detect obstructive CAD on a regional basis, and a global stress MBF value of <1.82 mL/g/min to accurately discriminate between obstructive three-vessel disease and CMD [57]. An alternative solution that is emerging is the use of newer contrast agents that show a linear dependency of signal intensity on concentration [58].

On the other hand, a recent systematic review summarizing other studies reported fairly overlapping values of myocardial perfusion reserve index (MPRI) for angina patients without obstructed coronary arteries and controls without CMD (mean MPRI ranged from 1.47 ± 0.36 to 2.01 ± 0.41 in patients and from 1.50 ± 0.47 to 2.68 ± 0.49 in controls without CMD). Therefore, no definite cut-offs/reference values could be provided. Probable reasons for this wide divergence include the inconsistency of the reference standard used; the researchers not always being blinded to the values of this reference standard; and the use of different types and doses of vasoactive agents to induce hyperemia [59].

Another challenge in perfusion imaging by CMR is the “dark rim artifact”, which not only affects visual image analysis, but can also significantly interfere with MBF quantitation as well. This subendocardial dark artifact is inherently because of the use of Gadolinium-based contrast agents [48].

### 5.2. Invasive Diagnosis

Invasive evaluation remains the clinical reference standard for diagnosing the different INOCA endotypes, and is as yet the only method to offer a comprehensive assessment of all aspects of microvascular dysfunction and spastic disease.

The invasive assessment starts, essentially, with a standard diagnostic coronary angiography to rule in/rule out obstructive CAD. This should include intracoronary physiological assessment (FFR/iFR) and/or intravascular imaging at any suspicion of a false negative angiography. Subsequently, a comprehensive physiological assessment is usually performed ad-hoc, but also can be staged on another session if the patient is inadequately prepared. The sequence of the different components of the assessment can differ among institutions [12]; our institutional protocol using coronary thermodilution, including a brief description of the discrepancies with the COVADIS protocol, is presented in Figure 1 and Figure 2.

The EAPCI consensus recommends glyceryl trinitrate (GTN), which has a short half-life during coronary angiography, based on the predication that a corrected TIMI frame count > 27 in the presence of GTN is suggestive of CMD [12,36]. In our perspective, since the resting caliber of the coronaries is very variable, and so is the response to GTN, all studies should be performed in the absence of GTN. To prevent radial artery spasm and pain, the guiding catheter should be advanced over a 4F pigtail or multipurpose catheter in order to reduce the vascular trauma. Mild sedation with benzodiazepines is, in general, sufficient. In the case of spasm, a femoral approach (or staged exam) should be preferred to administering medications that may interfere with vascular responses.

#### 5.2.1. Patient Preparation

All vasoactive medications and methyl xanthine-containing agents are withheld at least 24–48 h prior to the scheduled procedure (according to half-life). Smoking and short-acting nitroglycerin should be refrained from for at least 4 h before the procedure [34].

#### 5.2.2. Assessment of Endothelial-Independent Microvascular Function

This is done through the assessment of CFR (the ratio of hyperemic to resting blood flow), IMR (the product of intracoronary pressure times mean transit time, which is inversely proportional to flow) and HMR (the ratio of mean distal coronary pressure to flow velocity).

Absolute coronary flow is not measured in the clinical setting using standard methods, but either thermodilution or Doppler-based techniques are used to derive CFR. The most recently presented technique allows for measuring absolute flow and resistance, which offers clear advantages [60]. All techniques require the use of an agent to induce hyperemia (such as adenosine, papaverine or regadenoson) [61].

##### Thermodilution Technique

Principle: coronary blood flow is equal to vascular volume/blood flow velocity, calculated as mean transit time (Tmn) of a saline solution injected as a bolus at room temperature. Assuming a constant vascular volume, CFR calculation can be reduced to Tmn at rest/Tmn during hyperemia [4].

Method: A guidewire with both pressure and temperature sensors is positioned in a distal coronary artery (preferably the LAD). Resting mean transit time (Tmn) is then measured by at least three repeated injections of ~3 mL saline bolus at room temperature. The aortic pressure at the guiding catheter (Pa) and the distal coronary pressure (Pd) at the tip of the guidewire are recorded simultaneously (to derive the FFR). Subsequently, measurements are repeated during maximal hyperemia. Besides the calculation of the CFR (average Tmn rest/average Tmn hyperemia), the “index of microvascular resistance” (IMR) is calculated as Pd at maximal hyperemia multiplied by the hyperemic Tmn [62].

It is worth mentioning that women might have a shorter resting Tmn than men, i.e., higher resting coronary flow than men, which would yield a lower CFR in women for the same IMR [63].

##### Doppler-Based Technique

This technique is more demanding because it can be challenging to obtain a stable Doppler signal [34].

Principle and method: it has been shown that there is a negligible variation in epicardial vessel diameter/cross-sectional area in response to adenosine. Therefore, coronary flow velocity is used as a surrogate of flow. Accordingly, coronary flow velocity reserve (CFVR) is calculated as flow velocity during hyperemia/flow velocity at rest [4]. In addition, because microvascular resistance is the ratio between myocardial perfusion pressure (which can be reduced to Pd) and flow, the “hyperemic microvascular resistance” (HMR) is calculated as hyperemic Pd/APV, where APV is the average peak hyperemic velocity [4].

##### Continuous Thermodilution Method

Recently, a method based on thermodilution and the continuous infusion of saline at room temperature (rather than bolus injections) was developed to calculate the absolute coronary blood flow and the microvascular resistance in a simple, operator-independent manner, which was shown to be reproducible and safe [60,64].

##### Cut-Off Values

The EAPCI endorses a CFR cut-off of 2.0 for invasive thermodilution (in the presence of normal FFR/iFR/RFR) and/or IMR > 25, and a CFR cut-off of 2.5 for Doppler-based measurement [12,65]. Other authors showed that the optimal cut-off value of CFR to predict adverse events is 2.26 [66]. Studies demonstrating the prognostic value of Doppler-based CFR used different cut-offs, e.g., one group defined normal as a value ≥ 2.32, another used a value < 2.0 to define abnormal, and for yet another group the optimal prognostic cut-off value was 2.5 [67,68,69].

The normal range for IMR is < 25, based on three studies on healthy populations [70]. Different cut-offs for HMR were reported. An HMR value > 1.9 mmHg/cm/s independently predicted ongoing angina at short-term follow-up in one study [71]. The same value was associated with an increased risk of reversible myocardial ischemia as assessed by SPECT, even in the presence of collateral flow (independent of the FFR) [72]. Other studies showed that a cut-off of ≥ 2.5 offered the highest sensitivity and specificity for predicting CMD, as validated with CMR MPRI [73]. Of note, while the value of IMR and HMR is rarely discussed, the diagnosis of CMD only based on a pathological CFR in the absence of stenoses is controversial. Since CFR is a marker of the change in blood flow from rest to hyperemia, the pathological value might be due to a reduced maximum flow (i.e., impaired vasodilation) and an increased resting flow (i.e., vasodilation prior to administration of adenosine/papaverine). This latter condition is typical of high-flow states, such as hypertension, anemia, diabetes mellitus, etc., independently of an involvement of the microcirculation [74]. Particular care should therefore be taken in these contexts to avoid false positive diagnoses.

#### 5.2.3. Assessment of Endothelium-Dependent Microvascular and Macrovascular Function (Provocation Testing for Coronary Vasospasm)

Principle: in the presence of a healthy endothelium, ACh dilates the microvascular as well as the epicardial circulation, because nitric oxide is generated by the endothelium that acts on the surrounding vascular smooth muscle and overcomes the constricting action of the muscarinic receptors. In endothelial dysfunction, vasodilation is reduced, and vasoconstriction may even occur.

Method and interpretation: ACh is administered as a slow infusion for up to 3 min via the coronary guiding catheter, with increasing doses (full protocol described in Figure 2 and elsewhere [75]). Microvascular endothelial dysfunction is diagnosed if CBF increases < 50% from baseline, and/or ischemic ECG changes and pain occurs, in the absence of significant epicardial vasoconstriction (defined as > 90% diameter reduction). Another indicator of microvascular dysfunction is the development of a slow flow, i.e., reduced TIMI flow or increased TIMI frame count. On the other hand, vasospastic angina is diagnosed as defined by COVADIS, with the reproduction of the usual chest pain, ischemic ECG changes, and > 90% vasoconstriction on coronary angiography (even though reliance on angiographic stenosis alone does not warrant hemodynamic relevance, and given the poor correlation between angiography and ischemia, the spasm should be confirmed by the appearance of a pressure gradient as measured by a pressure wire left in place during the assessment) [37].

Apart from injecting nitroglycerin (GTN) if ischemia occurs during ACh provocations, the international microcirculation working group recommends that GTN should be delivered via a guiding catheter at the conclusion of the ACh infusions, to evaluate non-endothelial dependent epicardial smooth muscle (macrovascular) function, with a normal response defined as a > 20% diameter increase by QCA [75].

#### 5.2.4. Safety

The above invasive tests are not totally devoid of risk. However, it is argued that their addition does not significantly raise the overall procedural risk, particularly if balanced against the value of the gleaned information [76]. The most feared complication is spasm potentially leading to arrhythmia and cardiac arrest, or an infarction at the least. This complication is, however, extremely rare, and most cases of spasm are reactive to GTN or atropine (atropine being an antagonist of muscarinic receptors). Another possible complication includes coronary artery dissection with instrumentation, such as the infusion catheter or wire. The overall reported complication rate is 0.7–1.3%. Nevertheless, by paying attention to detail, ensuring careful monitoring, and following contemporary standardized protocols, potential complications can be mostly avoided or quickly reversed to avert serious harm [75,77]. Therefore, the EAPCI’s view is that the potential risk of such invasive assessment should be weighed against the benefit of the diagnosis for the individual patient, considering that so far there is no evidence base for an influence on prognosis, with only one pilot trial (CorMicA) that found a benefit in terms of symptoms and quality of life [12,78].

#### 5.2.5. Interpretation

Invasive assessment should yield one of the relevant endotypes described in Table 1.

## 6. Therapeutic Approaches and Strategies

Specific disease-modifying therapies remain an unmet need. However, there is now evidence that a stratified medical treatment based on the above comprehensive invasive assessment enables the marked and sustained improvement of symptoms and quality of life [78,79]. Randomized outcome trials are still needed.

### 6.1. Pharmacotherapy

A scheme of guidelines-recommended therapies is presented in Table 2.

#### 6.1.1. First Line (Conventional) Drugs

##### Microvascular Angina

Treatment should tackle the dominant mechanism [80]. In patients with abnormal microcirculatory vasodilation (reduced CFR or elevated resistance), and a negative ACh-provocation test, the underlying pathophysiology is mostly that of structural remodeling. Therefore, treatment in such case is directed towards counteracting these changes with statins, angiotensin-converting enzyme inhibitors (ACEi) or receptor blockers (ARB), and lifestyle changes, as well as lowering myocardial oxygen demand by beta-blockers and weight loss. A summary of studies on these drugs is provided in Appendix A.

However, for patients with microvascular spasm, treatment should be similar to that for epicardial vasospastic angina patients [81] (see below).

##### Vasospastic Angina

Whether epicardial vessels or microvessels (or both) are involved, the underpinning pathophysiology is over-excitability of medial smooth muscle cells and/or reduced nitric oxide production by injured endothelium. Therefore, the treatment should counteract the smooth muscle cell contraction using calcium channel blockers (CCB) (studies summarized in Appendix A) and nitrates, and replace nitric oxide with long-acting nitrates. In addition, improving the endothelial function should be attempted by controlling cardiovascular risk factors and lifestyle changes. Use of β-blockers as monotherapy should be avoided in patients with vasospastic disorders.

#### 6.1.2. Second-Line Drugs

##### Nicorandil

Nicorandil is a vasodilator agent with a combined action: A nitrate-like action (activation of guanylate cyclase) and potassium channel activation. Studies showing the benefit of microvascular angina are summarized in Appendix A. Although recommended by the Japanese Circulation Society for the treatment of vasospastic angina (Class IIa) [82], most supporting evidence is derived from the anecdotal success of intracoronary or intravenous bolus injection.

##### Ranolazine

Studies on ranolazine in CMD yielded conflicting results, but were generally to ranolazine’s favor, especially in patients with CFR invasively assessed to be < 2.5. Ranolazine is an anti-anginal agent that has no hemodynamic effect (i.e., no impact on heart rate or blood pressure). The specific mechanism of action has not been entirely elucidated. Based on preclinical studies, it inhibits the late phase of the inward sodium channel (late I_Na_) during cardiac repolarization, suppresses early afterdepolarizations, and reduces transmural dispersion of repolarization [83]. The reduced intracellular sodium concentration precludes intracellular calcium overload, which in turn improves diastolic mechanical dysfunction and inefficient energy consumption, and decreases micro-circulatory resistance by reducing wall tension and consequently the compression of intramural vessels [84]. In a pilot study, ranolazine was shown to improve myocardial perfusion in patients who had reversible perfusion defects on treadmill gated SPECT [85].

Additionally, ranolazine might have a protective effect during ischemia–reperfusion by reducing superoxide emission and reducing necrosis and apoptosis [86], and it appears to improve endothelial function [87] (Appendix A).

##### Ivabradine

Ivabradine is an established second-line agent for stable angina [88]. It selectively blocks the If-current of sinoatrial node cells, reducing sinus node activity. In contrast to β-blockers, it does not have a negative inotropic effect, nor does it cause vasoconstriction.

In patients with CMD, one study showed that it improved angina, whereas coronary microvascular function (in terms of CFR assessed by TTE) did not change, suggesting that symptomatic improvement is totally attributable to its classic heart rate-lowering effect [89]. However, in another study, ivabradine improved CFR invasively measured in non-culprit vessels of stable CAD patients, even when paced to a heart rate identical to that before initiating treatment, suggesting a genuinely improved microvascular function [90] (Appendix A).

##### Trimetazidine

Trimetazidine is known to reduce ischemia by blocking the beta-oxidation of fatty acids and thus favoring glucose oxidation. Studies investigating its role in CMD have had inconsistent results.

In a placebo-controlled double-blind study, 35 patients (mostly women) received trimetazidine 60 mg daily for two 4-week treatment periods. The study end-point was symptom-limited exercise testing. Trimetazidine prolonged total exercise time and time to 1 mm ST depression compared to placebo. In addition, the maximum ST depression was less with trimetazidine. Therefore, it was concluded that trimetazidine is beneficial in microvascular angina [91]. These results were confirmed by another similar-sized study [92]. However, in another study that compared trimetazidine and atenolol (see Appendix A), only atenolol improved symptoms, exercise performance, and TTE Doppler indices of diastolic function [93].

##### Xanthines

Xanthine derivatives (e.g., caffeine, aminophylline, paraxanthine, pentoxifylline, theobromine, and theophylline) have been reported to improve symptoms, but not necessarily ECG evidence, of ischemia in INOCA patients (Appendix A). Their mechanisms of action include the antagonism of adenosine-dependent vasodilation preventing coronary steal to non-dysfunctional regions, and a direct analgesic effect [94,95].

#### 6.1.3. Novel and Experimental Pharmacotherapies

Appendix A summarizes studies on novel drugs being investigated in the treatment of INOCA/ANOCA. Some previously tested drugs such as adrenoceptor-1 blockers showed no efficacy [96,97]. However, others, such as Endothelin receptor antagonists and Rho-kinase inhibitors, hold some promise.

### 6.2. Non-Pharmacologic/Interventional Therapies for Refractory Cases

Enhanced external counter pulsation (EECP), spinal cord stimulation (SCS), and most recently coronary sinus reducer (CSR) implantation, are non-pharmacologic alternatives to be used as a bail-out if pharmacotherapy fails [98]. Appendix A summarizes studies investigating these drugs.

## 7. Prognosis

Despite earlier studies reporting good prognosis [99], it was later shown that coronary endothelial dysfunction, in the absence of obstructive coronary artery disease, is associated with increased cardiac events, such as myocardial infarction, percutaneous or surgical coronary revascularization, and death [100,101,102,103]. However, the long-term prognosis of microvascular spasm might be better than that of epicardial spasm [104]. A relatively large multi-national study reaffirmed the substantial risk of MACE portended by microvascular angina [105,106].

Different invasive techniques with various reported CFR cut-offs depending on the method used predicted adverse cardiac events in INOCA/ANOCA patients [66,67]. In a multicenter study, the resistive reserve ratio, measured as CFR × (Resting Pd/hyperemic Pd), had an incremental prognostic value above CFR per se [107].

On the other hand, from a non-invasive perspective, PET-derived CFR < 2.0 predicted MACE (cardiac mortality, nonfatal myocardial infarction, late revascularization, and hospitalization for heart failure) in a study of 405 men and 813 women with no visual evidence of coronary artery disease [108].

Similarly, stress perfusion CMR-derived MPRI independently predicted MACE in a study involving 218 patients, even after adjusting for other possible predictors. These patients were followed-up for a median of 5.5 years for a composite MACE outcome defined as all-cause mortality, acute coronary syndromes, development of epicardial obstructive coronary artery disease, heart failure hospitalization, and non-fatal stroke. An MPRI threshold of ≤ 1.47 predicted MACE, meaning that patients underwent a 3-fold increase compared with patients whose MPRI was > 1.47 [109].

Of note, regardless of sex, CFR is a strong incremental predictor of MACE, after adjusting for clinical risk and ventricular function [108]. However, other studies point to a sex-difference in prognosis. For instance, a multicenter study from South Korea reported that a low CFR (≤2.0 by thermodilution) predicted MACE compared to a high CFR in men, but not in women [110].

Apart from vascular events, several studies have shown evidence of CMD in patients with heart failure with preserved ejection fraction (HFpEF), such that it might be a key pathophysiologic player [111,112,113,114,115,116,117]. Moreover, in patients without flow-limiting epicardial CAD, impaired CFR was independently associated with diastolic dysfunction and adverse events [118,119]. Furthermore, CMD has been shown to be associated with subtle alterations in systolic performance and combined measures of myocardial performance [120].

## 8. Knowledge Gaps and Limitations

A significant limitation of the studies investigating various therapies for INOCA is the adoption of diverse definitions of microvascular angina (heterogeneous inclusion criteria) and different endpoints (symptoms versus non-invasive or invasive assessment of the microcirculation) that seldom include hard end-points. Moreover, the majority did not rely on invasive assessment, mostly because they pre-dated the standardized definitions of CMD and vasospastic angina, which came out later. Another inadequacy is that most studies are small-sized and single-center.

Emerging attempts have been made to mend these deficiencies, exemplified by the WARRIOR trial (NCT03417388; estimated completion date: 30 December 2023) testing the effect of an intensive statin/ACE-I (or ARB)/aspirin treatment strategy; the iCORMICA (NCT04674449) and EXAMINE-CAD trials, testing the effect of a stratified endotype-specific treatment protocol, and the COSIMA trial (NCT04606459), testing the effect of the coronary sinus reducer in patients with IMR > 25.

## 9. Future Directions

Further adequately powered studies are needed to develop markers for risk-stratifying patients with ANOCA/INOCA, to determine who would benefit from more frequent monitoring or more intensive treatment. Studies to elucidate the molecular and cellular mechanisms underlying the different described endotypes are warranted. This would lend a better mechanistic understanding of the disease with the exploration of new therapeutic interventions that would target such mechanistic pathways. More evidence has to be generated to guide individualized and precision therapies.

## Figures and Tables

**Figure 1 biomedicines-09-01774-f001:**
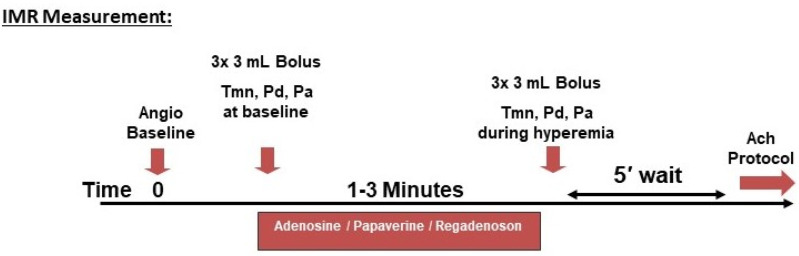
IMR measurement; University Medical Center Mainz institutional protocol for invasive assessment of INOCA/ANOCA.

**Figure 2 biomedicines-09-01774-f002:**
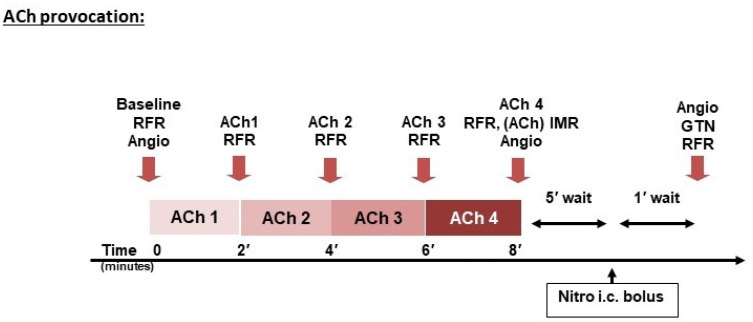
ACh provocation; University Medical Center Mainz institutional protocol for invasive assessment of INOCA/ANOCA.

**Table 1 biomedicines-09-01774-t001:** Schematic presentation of the different phenotypes of vasomotor vascular dysfunction.

Diagnosis	Mechanism	Definition
**Microvascular Disease**		
	Abnormally high microvascular resistance at rest	Coronary slow flow/Syndrome Y
	Impaired microvascular relaxation	With microvascular disease: IMR > 25 AND/OR HMR > 2.4 No clear conclusion about microvascular disease: CFR < 2.0 with FFR > 0.80 and/or resting indexes > 0.89
	Microvascular spasm	Angina during intracoronary infusion of acetylcholine with typical ischemic ST-segment changes, FFR/resting indexes normal AND IMR > 25 immediately after highest dose Ach
**Epicardial Disease**		
	Epicardial spasm	(1) reproduction of the usual symptoms AND; (2) ischemic ECG changes (1 mm horizontal or downsloping ST depression OR ST elevation OR T Wave inversion) AND; (3) >75% vasoconstriction on angiography AND FFR < 0.80 OR resting indexes < 0.89
	Obstructive epicardial disease	FFR < 0.80 Contrast FFR < 0.83 Resting indexes < 0.89

IMR: index of microvascular resistance; HMR: hyperemic microvascular resistance; FFR: fractional flow reserve; CFR: coronary flow reserve.

**Table 2 biomedicines-09-01774-t002:** A scheme of the different therapies based on phenotypes.

Endotype	Diagnosis: Coronary Vasomotion Disorder	Stratified Medical Therapy
Microvascular angina	IMR ≥ 25\(Microvascular resistance)	Baseline therapy: Consider aspirin, statin and ACE inhibitor therapy in all patients. PRN sublingual GTNAntianginal therapy 1st Line—Beta blocker (e.g., nebivolol 2.5 mg OD or carvedilol 6.25 mg BD uptitrated) 2nd Line—Calcium channel blockers (CCB) substituted (Non DHP e.g., verapamil 40 mg BD uptitrated)—where β-blockers are not tolerated or are ineffective 3rd Line—Add in therapy (avoid long acting nitrates) • CCB—DHP (e.g., amlodipine)—only for those on beta-blockers • Ranolazine (375 mg BD, uptitrated) Avoid long-acting nitrate unless previously established good response or co-existent epicardial spasm
	CFR < 2.0 (Coronary vasorelaxation)	
	Microvascular spasm to Ach (Propensity to microvascular constriction)	
Vasospastic angina	Epicardial spasm (>90%)	Baseline therapy: If atherosclerosis or endothelial impairment, aspirin, statin and ACE inhibitor should be considered. PRN sublingual GTN Antianginal Rx 1st Line—Calcium channel blocker (CCB)—e.g., verapamil 40 mg BD uptitrated 2nd Line—Add nitrate—e.g., PETN 50 mg BD-TID
Non-cardiac	Nil	Cessation of antianginal therapy. Stop antiplatelet and statin unless other indication. Consider non-cardiac investigation or referral where appropriate (e.g., psychological referral, gastroenterology)

## Data Availability

Not applicable.

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
