# Peer review of "Vasomotor Dysfunction in Patients with Ischemia and Non-Obstructive Coronary Artery Disease: Current Diagnostic and Therapeutic Strategies"

_biomedicines, 2021, doi:10.3390/biomedicines9121774_

Round 1

Reviewer 1 Report

The manuscript reviewed the ischemia with non-obstructive coronary arteries based on the invasive coronary angiography, focusing on the possible mechanisms and current therapeutic medicine. The topic is of significance. However, I would recommend the authors carefully read and rephrase the description for the readers to follow. For example:

  1. The journal is Biomedicines, but the footer shows J.Clin. Med. It looks to be confusing.
  2. The title seemed not to be rigorous; please consider rephrasing like “Ischemia with non-obstructive coronary artery disease: current diagnostic and therapeutic strategies.”
  3. Abstract, to allow stratifying, consider replacing “to stratify” with the -ing form.
  4. Introduction, Paragraph 1, consider changing the punctuations before and after the abbreviation, i.e., ------an anatomic strategy, i.e., computer tomography------ In addition, please cite the literature for the terms ANOCA and INOCA, and elaborate on them.
  5. Please clarify the phenotype and endotype for the reader to follow easily.
  6. Page 21, Paragraph 1, missing a period in the end.
  7. At the end of the manuscript, the authors addressed the limitation and the ongoing studies currently. It would be more helpful to add future directions or perspectives.
  8. There are numerous abbreviations and descriptions for the readers not to easily follow in the paper, such as SLE in paragraph 3.1; please spell it out when first shown in the manuscript; CA+ in tables 4, 6, &7, etc.

Author Response

The manuscript reviewed the ischemia with non-obstructive coronary arteries based on the invasive coronary angiography, focusing on the possible mechanisms and current therapeutic medicine. The topic is of significance. However, I would recommend the authors carefully read and rephrase the description for the readers to follow. For example:

  1. The journal is Biomedicines, but the footer shows J.Clin. Med. It looks to be confusing.

Reply: Thank you very much for this comment and apologies for the confusion. We corrected the footers and headers by placing the correct Journal title “Biomedicines” instead of “J. Clin. Med.”.

  1. The title seemed not to be rigorous; please consider rephrasing like “Ischemia with non-obstructive coronary artery disease: current diagnostic and therapeutic strategies.”

Reply: Thank you. Indeed the title you suggested describes the content of the manuscript more precisely and clearly. We therefore changed the title to match your suggestion.

  1. Abstract, to allow stratifying, consider replacing “to stratify” with the -ing form.

Reply: Thank you. We replaced the infinitive form “to stratify” by the gerund form “stratifying” to improve readability and understanding.

  1. Introduction, Paragraph 1, consider changing the punctuations before and after the abbreviation, i.e., ------an anatomic strategy, i.e., computed tomography------ In addition, please cite the literature for the terms ANOCA and INOCA, and elaborate on them.

Reply: Thank you for both comments. Firstly, for the abbreviation “i.e.” we added the proper punctuation for an interrupting word as you explained, by adding a comma before and a comma after the abbreviation. Secondly, we expanded the paragraph in the introduction that first mentions the terms ANOCA and INOCA in the review, so that the reader can easily follow, and added citations for the relevant literature (references number 4 and 5 in the resubmitted/revised manuscript).

  1. Please clarify the phenotype and endotype for the reader to follow easily.

Reply: Thank you. We agree this is a very important distinction, which was acknowledged by the EAPCI expert consensus document. Therefore, we expanded the first paragraph under the heading “2. Pathophysiology and endotypes” so that the reader can clearly make the distinction between both concepts.

  1. Page 21, Paragraph 1, missing a period in the end.

Reply: Thank you very much, a period was added at the end. Similar correction was made to the 2nd paragraph under the heading “6.1.2.2. Ranolazine”.

  1. At the end of the manuscript, the authors addressed the limitation and the ongoing studies currently. It would be more helpful to add future directions or perspectives.

Reply: Thank you for the suggestion. A new section was added at the end of the manuscript under the heading “8. Future directions”. This section briefly proposes different routes that future studies should explore.

  1. There are numerous abbreviations and descriptions for the readers not to easily follow in the paper, such as SLE in paragraph 3.1; please spell it out when first shown in the manuscript; CA+ in tables 4, 6, &7, etc.

Reply: Thank you. We made sure we spelled out any abbreviation on its first mention in the manuscript, including “SLE”. We replaced the abbreviation “CA” by “CAG” so that it refers uniformly throughout the manuscript to “coronary angiography”, and clarified what it means in the various table footnotes.

Reviewer 2 Report

This is a complete and outstanding review on this important topic. The paper is well organized and it is likely to read.

If I've to find a criticism this refers to the imbalance from previous therapies and new approaches: while the first are well known the new approaches should have more space and discussion. Nevertheless the paper is suitable for publication also in this form

Author Response

Thank you very much for your appraisal. Your comment is well-taken. In order to keep the review concise, we tried to briefly present the evidence-base for each therapy. For the more conventional therapies, there seems to be more published studies, albeit of small size and mostly single-center. This might have certainly contributed to this sense of imbalance. Besides, there are other previously published reviews which are more dedicated to the new therapeutic approaches rather than surveying the whole subject, e.g., the review by Rakhimov K. on “Non-pharmacological Treatment of Refractory Angina and Microvascular Angina”.

Reviewer 3 Report

Although we aknowledge that the topic of this review is relevant to cardiologists and clinicians in general, we think that the approach chosen by the authors, i.e. to provide a narrative, descriptive report of the existing information on the subject, does not reach the objective to provide an organised summary of the state of the art on the subject. Besides the pathophysiologic background of microvascular disease and vasospastic angina, the different diagnostic techniques and especially the therapeutical/pharmacological strategies are reported with too many details without providing to the reader the essential information.

We think that aa extensive major revision of the paper is needed to make it suitable for publication in Biomedicine journal : we suggest that the authors revise their results and produce a systematic review of the literature, include the methods  followed to retrieve the relevant articles, and compare the main results of diagnostic approaches and current therapies: this objective can be reached also by a limited number of more readable Tables. We suggest the authors to include also the adverse events in A/INOCA patients, i.e. myocardial infarction with normal coronaries and/or non obstructive CAD, which is a topic of great interest for cardiologists.

As a minor comment, several typos and unclear sentences are present along the manuscript (for example on pag. 2,8,13,18, etc)  and the legends of the Tables are missing or incomplete.

Author Response

Dear reviewer,

Thank you very much for your comments, which surely helped us improve the manuscript.

Your comment: Although we acknowledge that the topic of this review is relevant to cardiologists and clinicians in general, we think that the approach chosen by the authors, i.e. to provide a narrative, descriptive report of the existing information on the subject, does not reach the objective to provide an organised summary of the state of the art on the subject. Besides the pathophysiologic background of microvascular disease and vasospastic angina, the different diagnostic techniques and especially the therapeutical/pharmacological strategies are reported with too many details without providing to the reader the essential information. We think that aa extensive major revision of the paper is needed to make it suitable for publication in Biomedicine journal: we suggest that the authors revise their results and produce a systematic review of the literature, include the methods followed to retrieve the relevant articles, and compare the main results of diagnostic approaches and current therapies: this objective can be reached also by a limited number of more readable Tables.

Our Reply: Thank you for your comments. The details mentioned in the sections that address the pathophysiology, epidemiology and diagnostic approaches, are meant to introduce the reader who might not be familiar with the key aspects to the subject. We tried to simplify this section to avoid redundancy. In an attempt to help readers with a more thorough background to skim through familiar information more quickly, we placed the key messages at the beginning of each paragraph and divided paragraph 2 in smaller sections. As for the therapeutic/pharmacological strategies, we tried to do justice to the published evidence supporting/refuting each individual therapy (as commended by the other 2 reviewers) using a systematic review approach. This approach however increased the amount of information that needed to be entered in the tables (using a systematic review approach implies that all papers, not just a selection, is quoted). In order to simplify the manuscript, we relocated several tables (tables 3-11; some of which were expanded) to “supplementary material” (now supplemental tables 1-9), to improve readability such that key information stands out more clearly. Whenever the evidence was insufficient, or conflicting, this was mentioned.

We now also explicitly explain the systematic methodology used to retrieve the relevant articles in section 1.1. of the manuscript.

Your comment: We suggest the authors to include also the adverse events in A/INOCA patients, i.e. myocardial infarction with normal coronaries and/or non obstructive CAD, which is a topic of great interest for cardiologists.

Our Reply: Thank you very much for your comment, this is a key point. We added a separate section of the manuscript (7. Prognosis) to better explain which types of events the patients with A/INOCA and vasomotor disease experience and to provide a better understanding of the differential outcomes of this disease. If you agree, as reviewer 1 suggests, we would focus our work on "Vasomotor dysfunction in patients with Ischemia and non-obstructive coronary artery disease: current diagnostic and therapeutic strategies". Addressing the topic of the diagnosis of all possible causes of INOCA (including non-cardiac ones such as sepsis, anemia, etc.) would just confuse the reader.

Your comment: As a minor comment, several typos and unclear sentences are present along the manuscript (for example on pag. 2,8,13,18, etc) and the legends of the Tables are missing or incomplete.

Our reply: Thank you. We revised the language used, as you and reviewer 1 recommended, for spelling, grammatical, and punctuation mistakes. As recommended by reviewer 1 as well, we made sure that we explicitly spelled-out abbreviations/acronyms on their first mention in the manuscript. We also revised the table footnotes and made necessary amendments in several tables.

Round 2

Reviewer 3 Report

This revised version of the manuscript is significantly improved and suitable for publication in Biomedicines Journal.